# Cancer Prevention Literacy among Different Population Subgroups: Challenges and Enabling Factors for Adopting and Complying with Cancer Prevention Recommendations

**DOI:** 10.3390/ijerph20105888

**Published:** 2023-05-19

**Authors:** Lena Sharp, Nikolina Dodlek, Diane Willis, Arja Leppänen, Helena Ullgren

**Affiliations:** 1Regional Cancer Centre, Stockholm-Gotland, SE-10425 Stockholm, Sweden; 2Department of Nursing, Umeå University, SE-90187 Umeå, Sweden; 3Department for Oncology, University Hospital Center Osijek, 31000 Osijek, Croatia; 4Department of Nursing, Cyprus University of Technology, Limassol 3036, Cyprus; 5Nursing and Palliative Care, Faculty for Medicine and Dental Health, 31000 Osijek, Croatia; 6School of Health & Social Care, Edinburgh Napier University, Edinburgh EH11 4BN, UK; 7ME Head & Neck, Lung & Skin Cancer, Karolinska Comprehensive Cancer Center, SE-17176 Stockholm, Sweden; 8Department of Oncology and Pathology, Karolinska Institutet, SE-17177 Stockholm, Sweden

**Keywords:** cancer prevention, health literacy, cancer prevention literacy, public health, health inequalities, communication

## Abstract

It is estimated that 40% of the cancer cases in Europe could be prevented if people had better information and tools to make healthier choices and thereby reduce some of the most important cancer risk factors. The aim of this study is to gain knowledge and understanding about cancer prevention literacy among people with intellectual disabilities, immigrants, young people and young cancer survivors. In this qualitative study, we conducted six online focus-group interviews, including forty participants, to explore the cancer prevention literacy of four population subgroups and determine how cancer prevention recommendations according to the European Code Against Cancer (ECAC) were perceived. The analysis resulted in the following main categories: current health beliefs and their impacts on how the ECAC recommendations were perceived, communication strategies and sources benefiting or hindering cancer prevention information from reaching out, and how vulnerabilities in these subgroups impact cancer prevention literacy. To improve cancer prevention literacy in Europe, more attention is needed this topic to overcome barriers among different population subgroups. Recommendations include improved and adapted cancer prevention information, support to individuals, as well as societal support, such as easy-access screening and vaccination programmes and regulations related to tobacco, alcohol, and diet.

## 1. Introduction

Even if cancer mortality rates slowly decrease in parts of the world, the incidence is increasing, both in Europe and globally [1]. Exposure to modifiable cancer risk factors and access to screening and vaccination services are all impacting the cancer incidence rates. Cancer prevention has been proven to be more effective than curing it and is a more cost-efficient, long-term, cancer control strategy [2]. Despite this potential, cancer prevention has received little attention in most countries. Research shows that both the general public [3] and health-care professionals [4,5] have inadequate knowledge about the potential impact, which may explain the limited efforts seen so far.

Europe accounts for approximately 10% of the global population but also has 25% of the world’s registered cancer cases. It is estimated in Europe’s Beating Cancer Plan that 40% of the cancer cases in Europe were preventable [6]. However, extensive actions are required to improve both the information and the provision of resources to aid people in making healthier choices. Without these actions, the cancer mortality within the European Union (EU) is expected to increase by more than 24% by 2035, making it the leading cause of death across the continent [6].

Cancer prevention could potentially be improved considerably by raising awareness and addressing modifiable risk factors of cancer, such as tobacco and alcohol consumption, lack of physical activity, obesity, unhealthy diet, extensive sun exposure and exposure to pollution [1,2]. Improved cancer prevention involves modifying these unhealthy behaviours, as well as proposing recommendations on breastfeeding and taking part in cancer screening and vaccination programmes.

The “European Code against Cancer. 12 ways to reduce your cancer risk” is an initiative by the European Commission, developed by the World Health Organisation’s (WHO) International Agency for Research on Cancer (IARC) [7]. The European Code against Cancer (ECAC) aims to inform the public of the various actions they can take for themselves or for their families to reduce their cancer risk. The code includes 12 evidence-based recommendations, for example, taking measures to reduce weight, and is available in the 23 E.U. languages. Although the ECAC is a comprehensive tool aimed at communicating about cancer prevention, the general public’s familiarity with the code has been reported to be low [8].

Inequalities in both survival rates and other cancer outcomes have frequently been reported both within and between countries. Cancer risk factors are impacted by socioeconomic factors [9,10,11,12,13,14] and research has shown that socioeconomic inequities influence exposure to risk factors and access to screening and other preventive services, as well as diagnostics, cancer treatment and palliative care [10,15].

Health literacy is the degree to which people have the ability to use health information (rather than just understanding it) to make health-related decisions and take actions for themselves and others [16]. The WHO states that [17] raising health literacy among the most disadvantaged population subgroups is crucial to realise the sustainable developments goals set by the United Nations [18]. Low health literacy is connected with cancer misconceptions, low perceived control over risk factors, fatalistic beliefs about cancer [19] and low digital health information-seeking behaviours [20]. In a report by the European Commission, some population subgroups (especially people with intellectual disabilities (IDs), immigrants and young people) were highlighted as being in need of special attention regarding cancer prevention interventions to address current inequalities [9]. However, some of these groups have been described as difficult to reach with health promotions [10].

In this study, the term “cancer prevention literacy” is used to illustrate the specific health literacy aspects of cancer prevention.

Inequalities regarding access to prevention and screening services, as well as cancer care, may adversely affect underserved and/or unrepresented minority groups, for example, people with IDs or immigrants [13]. Research indicates that people with IDs have poorer general health compared to the general population. Their uptake of cancer screening is lower, and they have a higher risk of certain types of cancer [21,22]. Cancer incidence among people with IDs is increasing, yet little is known about cancer prevention literacy among this group [23].

There is evidence of lower general health and increased cancer risk among immigrants, refugees and asylum seekers [24]. Communication barriers and poor accessibility and attendance to cancer screening are contributing factors, in addition to health illiteracy and low education and income levels [14]. A better understanding of cancer prevention illiteracy among this group could positively impact any future prevention initiatives.

Other groups often reported as having low cancer prevention literacy are younger generations [25]. Early adulthood, for many, includes several changes and challenges, including milestones such as moving away from your parents, getting a job and becoming a parent yourself. Addressing certain behaviours and exposures during this period in life could severely impact future cancer risks. One dilemma, however, is the perception of these risks being seen as unrealistic or irrelevant by some young people. Better understanding of cancer prevention literacy among this group could improve and better tailor future cancer prevention initiatives.

For young cancer survivors, cancer treatment has been a major part of their life for some time. When treatment ends, routines, emotions and priorities may change whilst adjusting to life after cancer. Living with the risk of relapse, long-term complications and/or secondary cancers may affect experiences and implicate different perspectives, especially in comparison with other young people who do not have these experiences [26]. Young cancer survivors’ cancer prevention literacy is important to understand to be able to better address tertiary cancer prevention initiatives. However, little is known about facilitators and/or barriers of health literacy for this group.

The aim of this study is to gain knowledge and understanding on the subject of cancer prevention literacy among four population subgroups (people with IDs, immigrants, young people and young cancer survivors). Focusing on the cancer prevention recommendations in the ECAC (fourth edition), the aim was also to explore the challenges and facilitators of adopting and complying with evidence-based cancer prevention messages among these subgroups.

## 2. Materials and Methods

### 2.1. Study Design

We used a qualitative research design to explore cancer prevention literacy, including how the recommendations in the 4th edition of the ECAC were perceived by four different population subgroups.

### 2.2. Context

This study is a part of PrEvCan, cancer prevention across Europe (https://cancernurse.eu/prevcan/ accessed on 1 December 2022). PrEvCan is a large-scale cancer prevention program and is running from 2022 to 2023. It was initiated by the European Oncology Nursing Society (EONS) together with the European Society for Medical Oncology (ESMO) as a key partner. More than 65 other international and national campaign partners are involved. PrEvCan includes a cancer prevention campaign, education and research activities. The knowledge from this current study will be used to inform both the campaign and the education activities. An evaluation of the PrEvCan project is planned for 2024.

### 2.3. Data Collection

We conducted online focus-group interviews (FGIs) using the Microsoft Teams © software, with participants representing the four different population subgroups (described below). Each FGI was led by a moderator (last author, H.U.) and either one or two assessors (author L.S. for all six FGIs, and author N.D. for FGIs 1 and 2), who were actively listening, asking prompting questions, as well as summarizing the FGIs. These three authors have previous experience with FGI studies. Field notes were taken by both the moderator and the assessors. FGIs 5–6 (including participants with IDs) also included supporting staff (Table 1).

An interview guide with open-ended questions was developed by the research team and used as a base for the FGIs (Figure 1).

For reference, a description of the 12 recommendations in the ECAC 4th edition was shared on the computer screen. This description was adapted for the FGIs to include people with IDs (slides were prepared with only 2 of the 12 ECAC recommendations showing at a time to reduce the amount of text, and ECAC pictograms/illustrations were used to add clarity). The each FGI included between two and eight participants, and 90 min were allocated for each FGI. The FGIs were conducted in either English or Swedish (Table 1), filmed and then transcribed verbatim by an experienced research assistant. Focus groups were used as they enabled collaboration and have previously been used with hard-to-reach communities.

The concept of information power [27] was used during the planning and data collection phases to guide adequate sample sizes and robustness of the data. This concept suggests that the size of the sample can depend on stakeholder knowledge about the topic under study.

### 2.4. Recruitment and Participants

Separate online FGIs were conducted, with participants representing the four subgroups (see the background and Table 1). We strove to recruit a heterogenic sample from different geographical areas, as well as gender-mixed groups. Participants were recruited via gatekeepers (contact people) from the involved organisations (see below). For confidentiality purposes, the only personal information collected from each participant was their first name and country. All names of participants or organisations were changed in the transcripts, and only anonymised data were stored under the general data protection regulations of the lead researchers.

### 2.5. Subgroup A: Young People without Personal Cancer Experience (Aged 18–29, Focus Group 1 and 2)

These participants were recruited via the EONS Young Cancer Nursing network, which includes cancer nurses (aged 20–35 years old) from 15 different European countries. Their role in this study was to recruit participants from their personal social networks (persons aged 18–29 years old, without any personal cancer experience). None of the participants were known by any of the authors.

### 2.6. Subgroup B: People with an Immigrant Background (Focus Group 3)

These FGI participants were recruited via a network (organized by the Regional Cancer Centre Stockholm-Gotland) of peer advisors active in the multicultural areas of Stockholm and the region of Sörmland, Sweden. Peer advisors are engaged citizens from different age groups, working on a voluntary basis to reduce health divides. All peer advisors participating in this FGI had immigrated to Sweden from other countries and acted as peer advisors in multicultural communities. They actively spread/share information on cancer prevention and healthy lifestyle habits in their communities via public spaces/activities, schools and clubs. The coordinator of this network (who had no involvement in the study) approached the peer advisors, informed them about the study and also recruited participants.

### 2.7. Subgroup C: Young Cancer Survivors (Aged 18–29, Focus Group 4)

This group was recruited through the pan-European advocacy organisation, Youth Cancer Europe (YCE) (https://www.youthcancereurope.org/, accessed on 1 December 2022). As with the recruitment process for subgroup A, this network’s role was to inform and recruit participants from their membership (persons aged 18–29 years old, with personal cancer experience and from different countries across Europe).

### 2.8. Subgroup D: People with Intellectual Disabilities (IDs), Focus Groups 5–6

These FGIs were arranged in collaboration with advocacy groups in both Sweden and the United Kingdom (UK). The Swedish National Association for People with Intellectual Disability (https://www.fub.se/ accesses on 10 November 2022) is an advocacy organisation working to enable children, young people and adults with an intellectual disability to live a good life. FAIR (Family Advice and Information Resource) (https://www.fairadvice.org.uk accessed on 10 November 2022) is a third-sector, free, confidential information and advice service for people with learning disabilities, their families, supporters and carers in Edinburgh, UK. Staff and volunteers from both organisations recruited participants for the FGIs, and some members were also present to provide support and clarification to the participants if needed.

Verbal and written study information was provided to potential participants, via the involved organisations (described above) who acted as gatekeepers. All participants were reminded that taking part was voluntary and they could withdraw at any time, even during the FGI. Once a person had agreed to participate, they received a meeting link for the planned online FGI from the respective organisation. At the start of the FGIs, each participant was asked by the moderator whether they had received the study information. The FGI moderator repeated the aim and process of the study, and if the participant had any questions, highlighted that the participants had the option to opt out without explanation, if preferred. Once the participant confirmed and had no further questions, they were asked for verbal consent for their participation.

### 2.9. Ethical Considerations

All participants provided their informed consent (verbal and written) before they participated in the study. The researchers were aware that discussing cancer could potentially be stressful and worrying. It had the potential to bring up previous negative experiences from cancer diagnoses and treatments among the young cancer survivors. For all participants, the FGIs had the potential to cause stress related to their own previous cancer experiences or the experiences of others, such as family members and/or friends. Discussing lifestyle issues also had the potential to be taxing, adding feelings of guilt and shame about the participant’s own lifestyle choices and/or those of their family. To reduce these risks, all participants were offered support after the FGIs. This was either in the form of peer support from advocacy groups or professional support from a cancer support team (specialist cancer nurses with degrees in counselling).

The authors had no direct contact with the participants before or after the FGIs. An honorarium of EUR 250 was paid to each participant to cover any costs related to travel or taking time off work or studies. Incentivizing participation can be controversial, but it was not believed to be disproportionate to what was being asked of the participant. The honorariums were paid by either the Regional Cancer Centre Stockholm-Gotland or EONS to the involved organisations, who in turn compensated the participants.

### 2.10. Analysis

For analysing the FGI data, framework analysis (FA) was used. This qualitative analytic approach was developed by Ritchie and Spencer [28,29] and has been described as being well-suited for assessing policies among the people they affect [30]. In the FA, data are sifted, charted and sorted in accordance with key issues and themes/categories. The following five analytic steps were used: familiarization; identifying the thematic framework; indexing; charting; and mapping and interpretation (described below). These phases enabled the researchers to understand and interpret the data, moving from descriptive data to conceptual explanations. The authors met repeatedly in different constellations to discuss the ongoing analysis.

### 2.11. Familiarization

After conducting the FGIs, three of the authors (H.L., L.S. and N.D.) listened to the recordings, read the transcripts and discussed and compared each author’s field notes to familiarize themselves with the data. H.L. and L.S. had previous experience analysing qualitative data using FA.

### 2.12. Initial Framework

In FA, the researchers often begin the analytic process with an initial framework that is based on what is already known and/or what appears mostly relevant. In the current study, we developed an initial framework based on the following: the previous literature related to cancer prevention literacy, the field notes taken during the FGIs and the first stage of the analytic process (the familiarization). The initial framework included the following six themes: communication strategies that would benefit or hindering; perceptions of the ECAC recommendations; health beliefs; sources to trust/mistrust regarding health matters; vulnerability; and miscellaneous (where potentially interesting findings were listed that did not fit into any of the other initial themes) (Figure 2).

### 2.13. Indexing Data

The next step was to apply the initial framework to the data, with all textual data being pasted into the framework and organised as a matrix. Microsoft Excel © was used to organise the data. Notes were made with references to the FGI number and participant to facilitate tracing. We also made notes related to both the initial framework and possible final categories, which were repeatedly discussed within the research group (all authors).

### 2.14. Charting

Data were revised in the matrix with textual data and were sorted and shifted. Pieces of data indexed in the previous stage were arranged in charts. During this phase, the initial framework was amended, and the following three main categories were identified: Current health beliefs and their impact on how the ECAC recommendations were perceived, Communication strategies and sources benefiting or hindering cancer prevention information from reaching out,, and How vulnerabilities in the subgroups impacts cancer prevention literacy, Figure 2.

### 2.15. Mapping and Interpretation

This stage included interpretations of the dataset, identifying patterns and associations. The research team (all authors were involved here) used the matrix to find explanations and make interpretations, based on the aim of the study and the question guide (Figure 1).

## 3. Results

In total, forty participants were interviewed in six online focus groups (ranging from two to eight participants per FGI) during August and September 2022. The participants came from 13 different European countries (participants in subgroup B all lived in Sweden but had immigrant backgrounds). Most of the participants were women (*n* = 23, 58%) and the mean length of the interviews was 99 min (min–max 71–107). An overview of the FGIs is presented in Table 1.

The findings are presented below in the following main categories: current health beliefs and their impact on how the ECAC recommendations were perceived, communication strategies and sources benefiting or hindering cancer prevention information from reaching out, and how vulnerabilities in the subgroups impacts cancer prevention literacy. The results are illustrated with key quotations from the participants (in italics). Reference to the FGI number and the participant (P) are provided for each quote. Explanatory words, if needed by the authors, are inserted in the quotes in brackets. An overview of the analytic process and the final categories and subcategories are presented in Figure 2.

### 3.1. Current Health Beliefs and Their Impact on How the European Code against Cancer (ECAC) Recommendations Were Perceived

During the analysis, data from two of the themes (perception of the ECAC and health beliefs) in the initial framework (Figure 2) were further developed and merged into this category. We found that a strong desire to live a healthy life was a common denominator among the four subgroups, which was highlighted and described in the following subcategories: reflections on the European code Against Cancer (ECAC) recommendations and misconceptions about cancer risk factors.

### 3.2. Reflections on the European Code against Cancer (ECAC) Recommendations

During the FGIs, the 12 recommendations in the ECAC were discussed. Although some of the recommendations were perceived as logical and were well known (and therefore easier to consider), others were more unfamiliar and/or abstract. Participants in all six FGIs expressed general views of the ECAC recommendations as being overwhelming and therefore difficult to seriously grasp.

The focus of the ECAC is on what individuals can do to reduce their cancer risk, whereas the participants tended to reflect on a multitude of reasons as to why they would not follow the recommendations. A deeper understanding of both personal circumstances and factors impacting willingness, readiness and even resistance in making decisions seems to be required to better understand what people need to follow through with lifestyle changes. In all FGIs, the participants expressed a lack of societal support to aid these lifestyle changes.

It became apparent that these individual recommendations need to be combined with clear support structures (both on an individual and a societal level), and that these structures may vary between individuals and population subgroups. One woman with an ID described herself as being well aware of the health risks associated with overweight and obesity but struggled to be physically active and was therefore overwhelmed.


*I would have loved physical activity if it hadn’t been so much pressure about being…. It is dangerous to be overweight. It is of course dangerous, and it is dangerous to get cancer, you may get cancer by everything, it feels like, the way people talk today.*
(FGI5, P5)

Smoking as a risk factor for cancer (ECAC recommendation #1) was expressed as being well known among all subgroups, even if it was not always viewed as a realistic health threat among the younger participants (subgroup A).

The awareness of second-hand smoking (SHS) as a cancer risk factor (ECAC recommendation 2) varied both between and within the subgroups. Although some participants responded as being well aware of the health risks related to SHS, others were not, specifically as being a risk factor for cancer. Several of the participants (mainly subgroup D, people with IDs) raised issues with complying with this recommendation, demanding stronger legislations/bans (including firm consequences for noncompliance) against smoking in public places, residential homes and workplaces. In subgroup B (immigrants), questions were raised on the SHS risks from e-cigarettes and water pipes (also referred to as hookahs). They also expressed struggles in stopping their teenagers from both smoking and being exposed to SHS. The young cancer survivors (subgroup C) appeared to be well aware of the SHS risks, with some participants describing their struggle to comply because most or all of the people in their networks were smokers.

Most participants expressed their awareness of the importance of maintaining a healthy body weight and diet (ECAC recommendations 3 and 5), though some voiced that they were unaware of the related cancer risks. ECAC recommendation 3 (take action to be a healthy body weight) was questioned by some participants, with the phrase healthy body weight being considered unclear and unspecific.

Specifically for ECAC recommendation 5 (eating healthy), many of the participants explained difficulties in complying. Particularly among participants with IDs (subgroup D), the explanations given involved difficulties in knowing what to cook and understanding content labels when shopping, as well as a lack of support from staff members in their homes. Some participants also described being on medications that can increase body weight, which added to the complexity.

In general, most of the participants expressed an awareness of the health benefits of being physically active. However, some were unaware of the cancer risks associated with inactivity (ECAC recommendation 4).

Some of the participants were unaware of alcohol being a risk factor for cancer (ECAC recommendation 6). This recommendation was perceived as one of the most clear and understandable in the ECAC, even if some of the participants questioned the term “limit” and requested more specific instructions.

### 3.3. Misconceptions about Cancer Risk Factors

In general, the ECAC recommendations 7–12 (exposures to sun, radon and cancer-causing substances at work, as well as recommendations related to vaccinations, screening, breastfeeding and hormone therapy) were lesser known among the participants. We also found several misconceptions and myths in relation to these recommendations.

Most participants described themselves as being aware of sun exposure as a risk factor for skin cancer, but we also found misconceptions. In the FGI among immigrants (FGI 3), one of the participants described Sweden as being closer to the sun compared to other countries. Therefore, sun exposure would be particularly dangerous here, regardless of skin type.


*I have noticed and also read that in my homeland, the sun doesn’t affect our bodies, but here in Sweden, the sun is much, much closer. Many people don’t know about this.*
(FGI 3, P7)

The recommendation that seemed hardest for most of the participants to comprehend was ECAC recommendation 7 (on radon levels in peoples’ homes). Several misconceptions were described. Some participants described this risk factor as being related to exposure from using microwave ovens, mobile phones, web cameras, Teflon pans for cooking or having electric radiators in the home.

We also found misconceptions related to alcohol. Some participants believed in other health benefits of alcohol and therefore did not comply with this recommendation. This was mainly discussed among participants with immigrant backgrounds (subgroup B, FGI 3). A dialogue between two of the participants in this FGI illustrates this:


*I have read about red wine. It helps your heart and is good for your blood.*
(P7)


*I have noticed that a glass of wine helps his [the husband’s] diabetes, but not beer.*
(P3)


*So you mean that it helps to lower the blood sugar?*
(P7)


*Yes, it helps.*
(P3)


*I didn’t know that before.*
(P7)

Cancer risks in the workplace (ECAC recommendation 8) were generally described as being difficult to understand. Several participants expressed uncertainty on how to comply with this recommendation. They referred to trusting employers, trade unions and/or national authorities to handle the information and inform them of any cancer risks in the workplace. Others, however, expressed concerns about these potential risks because it was unclear to them what could be done.

We also found misconceptions related to ECAC recommendation 10 (on breastfeeding and hormone replacement therapy). This recommendation explicitly describes that breastfeeding reduces the mother’s cancer risk. Despite this, participants in several of the FGIs expressed their lack of awareness and were also unsure whether the risk reduction was related to the mother or to the baby.

We also found both concerns and misconceptions related to ECAC recommendation 11 (take part in vaccination programmes). Some of the participants in FGI 3 (immigrants) expressed a reluctance towards vaccinations in general. They described how many of the people they interact with, in their role as peer advisors, were reluctant to let their children be vaccinated against human papilloma virus (HPV). It was believed to cause infertility and/or encourage young people to be sexually active. The fact that this recommendation only mentioned girls also raised questions and assumptions that HPV vaccinations would be risky for boys and were therefore better avoided.

Recommendation 12 (take part in organised cancer screening programmes) also caused concerns and misconceptions among some participants. One of the more alarming concerns here was described by the peer advisors among the immigrant group (FGI 3), who explained that some women did not take part in gynaecological cancer screening due to a fear of exposing their genital mutilation and a belief of legal consequences (for example, losing their residence permit).

### 3.4. Communication Strategies and Sources Benefiting or Hindering Cancer Prevention Information from Reaching out

This category mainly consists of data from the following two themes in the initial framework (Figure 2): Communication strategies that would benefit or hinder and Miscellaneous.

In the FGIs, we asked the participants what they believed was important when communicating information pertaining to cancer prevention and how the dissemination of this information could be maximized. The results are presented in the following subcategories: Nuances in cancer prevention communication and Trusted or mistrusted sources of communication.

### 3.5. Nuances in Cancer Prevention Communication

Encouraging communication, focusing on positive actions rather than what to avoid, was especially requested by the younger participants (subgroups A and C) and to some extent by the participants in the other subgroups). One young man without personal cancer experience described trying to live a healthy life but was overwhelmed by the ECAC recommendations. He explained that he understood that these recommendations need to be clear but was provoked by what he perceived as being just a list of strict recommendations on what to avoid. He expressed a preference to more proactive, positive communication, where the risk factors were also explained more thoroughly.


*… to me if comes across as quite harsh, but I also accept this as a truth. Does it mean that I will automatically get cancer? [if not complying with the ECAC recommendations]. It feels unbelievable in a sense, when a statement is that short and that bold…. It feels apocalyptic in a sense.*
(FGI 1, P1)

He continued to reflect and concluded that if cancer prevention recommendations were communicated in such a way that clarified the gains of choosing to comply, he would be more inclined to follow them. He exemplified what he meant, related to smoking (ECAC recommendation 1).


*If you don’t smoke, it increases your… whatever… it gives you a positive sense of belief instead.*
(FGI 1, P1)

Similar perceptions came up in other FGIs. Being told what to avoid, especially if communicated by using a long list of recommendations, was believed to build resistance and contribute to feelings of irrelevance among the younger population. Comparisons were made by some participants on communication related to the COVID-19 pandemic, as this was also seen as irrelevant to many young people; therefore, recommendations were not followed.

It was also suggested that communication should focus on general aspects of health to better reach out with cancer prevention messages. For example, communicating health benefits of minor and thereby achievable changes in lifestyle or promoting easily accessed, health-promoting group activities were suggested.

Tailored communication to specific subgroups was requested, mainly from subgroups B (immigrants) and D (people with IDs). The peer advisors in FGI 3 explained that many people in their communities have limited knowledge about cancer and are unaware of some of the risk factors. Therefore, communication in different languages was described as essential. Some of the participants in this FGI also highlighted the importance of separate communication for men and women among some population groups. Furthermore, different ways of communication were requested because literacy is a problem in some of the communities. Here, verbal communication by people with culture-specific knowledge was recommended. In the two FGIs among people with IDs (FGIs 5–6), the participants also requested adjusted communication, such as using more illustrations and basic language, giving practical advice and information on where to access support.

Another suggested communication strategy was to integrate cancer prevention communication into situations where other health-related topics are discussed, for instance, during health check-ups with a school nurse, general practitioner or dentist.

Interestingly, the participants in subgroup A (young people without personal cancer experience) explained that cancer risk is often viewed as irrelevant when you are young. The participants in subgroup C (young cancer survivors) confirmed this general view among younger generations but also explained a different view, related to their own cancer experience. One young cancer survivor illustrated this with the following:


*That’s not how I think about it, mainly because of my past and stuff, but I really understand that our generation are like…. I really don’t care. It is not happening now, so it is never going to happen.*
(FGI 4, P2)

### 3.6. Trusted or Mistrusted Sources of Communication

In several of the FGIs, several sources for cancer prevention communications were discussed. Social media communication was explained as being preferred by many of the younger participants, but they also highlighted the importance of the trustworthiness of the sources. Participants in FGI 2 (young people without personal cancer experience) described difficulties in navigating communications related to health in general. One young woman explained receiving conflicting information related to nutrition, which caused confusion.


*There is a lot of misinformation on the internet that is available to all of us and then some say this about nutrition and someone else say something completely different. We don’t know what to believe actually. We don’t know what is good for us.*
(FGI 2, P4)

She continued to express concerns over the difficulties of gaining trustworthy information and explained that sometimes she could not even trust the sources that she normally would, such as her parents.

Social media influencers were suggested as being possible sources for cancer prevention communication, but it was pointed out that not all influencers can be trusted, as specific expertise is essential. Another young participant in FGI 2 explained how it can be difficult to navigate social media communication.


*You need to find someone who is an expert in the field and not someone who is followed by many, which might be hard. …sometimes people who are famous are communicating about things that they don’t know anything about… you need to find a group of experts.*
(FGI 2, P1)

We also found different preferences related to both cultural and individual factors. Some participants preferred communication via social media influencers or Facebook groups (mostly younger participants), others highlighted teachers, nurses and physicians as the most trusted sources, whereas others stated that communications from national authorities were most trusted. Several of the young cancer survivors (FGI 4) explained how the most trusted sources of any health-related information were the cancer specialists they met during their cancer treatments and follow-ups. They explained how challenging it could be for them to gain trusted information once the follow-up period was completed.

### 3.7. How Vulnerability in the Subgroups Impacts Cancer Prevention Literacy

This category mainly consists of data from the following two themes in the initial framework (Figure 2): *vulnerability* and *miscellaneous*. Each of the subgroups gave examples of both general and specific vulnerabilities that are believed to impact cancer prevention literacy. The results in this main category are presented in the following three subcategories: Vulnerabilities due to cultural and language issues, vulnerabilities due to disabilities or age and Vulnerabilities due to socioeconomic status.

### 3.8. Vulnerabilities Due to Cultural and Language Issues

Several aspects of vulnerability related to cultural and language issues were raised. Some of the young participants without personal cancer experience (subgroup A) explained cultural expectations related to unhealthy food and alcohol in some countries as contributing to vulnerability.

The participants with immigrant backgrounds explained that, sometimes, low attendance rates in cancer screening among this group are related to a lack of information, language barriers (not understanding the written invite and/or the potential benefits) and culture aspects (e.g., it is taboo to undress in front of medical staff unless critically ill or to expose genital mutilation). One of the participants explained how immigrants may be particularly vulnerable as they often, in her experience, not only lack information but also lack access to important information.


*Swedish people, they know a lot. They always have access to information, but foreign people are far less informed.*
(FG3, P4)

Another example of cultural vulnerability raised by the immigrant group was having to exclude yourself from important social events if striving for a healthy lifestyle. For instance, events including unhealthy foods and/or water-pipe smoking.

### 3.9. Vulnerabilities Due to Disabilities or Age

Vulnerabilities due to physical and mental disabilities were frequently mentioned, particularly among the participants with IDs. Physical disabilities were described as negatively impacting accessibility of health-care settings in general, as well as cancer-screening services. One participant in the ID group described being aware of the increased cancer risk for this subgroup. Despite this, she was not able to attend cancer screening due to a lack of support, knowledge and resources (e.g., being excluded from breast-cancer screening because supporting staff were unable to accompany her). Similar examples of vulnerabilities were raised by the same subgroup regarding their attempts to maintain a healthy diet, be physically active or avoid exposure to SHS.

Some participants stressed that the difficulties with keeping a healthy diet were because the essential motivational support was lacking. One participant preferred to cook and eat healthy food but described the lack of support from staff as stopping her.


*I tried to explain that I wanted to cook and eat healthy, but the staff asked if I really had to cook. Can’t you make something easy and quick from the freezer instead? So I did. They didn’t encourage me to be healthy at all.*
(FGI 5, P7)

Some positive examples were also raised in this group, describing situations in which the staff and group members worked together towards promoting health. Positive reassurance and group activities were described as reducing the vulnerability among this group. Examples were given relating to both healthy diets and physical activities.

The young cancer survivors also raised specific aspects of vulnerability that were related to both their cancer experiences and to their ages. One young woman described feelings of vulnerability, of being unable to avoid SHS after her treatment and of not knowing how to raise the issue amongst her friends, all of whom were smokers.


*I think every organ in my body and every cell fight it so much so I will be healthy… after all that I have been through. So I’m trying to explain my point of view to my friends… if they would understand maybe they would leave it and emphasize with me. But I don’t know how to say it and how they should take it. Maybe it is not possible.*
(FGI 4, P5)

### 3.10. Vulnerabilities Due to Socioeconomic Status

Aspects of vulnerabilities were also described as being related to socioeconomic status, especially among subgroups B (immigrants) and D (people with IDs). Contributing factors were explained as being the high price of healthy food and lack of education among this group. Many of the participants expressed a need for better support from society to aid with individual cancer prevention recommendations. Several suggestions were raised, such as easy-access, health-promotion programmes for people with IDs, better education for supporting staff, better compliance with smoking regulations to avoid SHS exposure, and improved availability for cancer-screening and vaccinations services, as well as improved information, focusing on the benefits of cancer prevention. Issues were also raised concerning support regarding healthy eating, as unhealthy food was seen as being more accessible and having a lower price.

## 4. Discussion

In this study, people representing different European population subgroups discussed how they perceived the cancer prevention recommendations in the ECAC, as well as the barriers and enabling factors related to cancer prevention literacy. To our knowledge, this is the first study wherein representatives from different groups (often described as important but difficult to reach with cancer information) shared their experiences in relation to the ECAC recommendations. The rich data include important knowledge that could improve cancer prevention literacy, not just specifically within each of the subgroups, but also in general.

Overall, the ECAC was believed to provide important information, as some of the risk factors were unknown. However, participants in all subgroups expressed that the recommendations would reach further and have a greater impact if there was a clearer focus on the health benefits and gains. The 12 ECAC recommendations are backed up with additional information, as well as scientific evidence on the IARC web pages. However, few people with low cancer prevention literacy are likely to find this information. Karasiewicz et al. [31] evaluated the ECAC in a study among the general public in a rural area of Poland and found knowledge gaps. A survey including over eight thousand respondents from eight European countries confirmed the low level of awareness of the ECAC amongst the general public [8]. Even if higher levels of awareness could be achieved, it would not be enough to improve cancer prevention literacy, specifically among vulnerable subgroups.

One important finding in the current study was the misconceptions involving cancer risk factors, mainly regarding ECAC recommendations 7–12. We also found misconceptions relating to alcohol (ECAC recommendation 5), which was particularly concerning given that these participants (FGI 3) function as peer advisors for cancer prevention in their communities. This highlights the importance of an adequate education for peer advisors. Misconceptions about cancer risk factors have previously been described [15,32,33]. Fleary et al. [34] found in their survey that cancer misconceptions were more common among participants with a lower level of health literacy. Even if the authors of this study concluded that health literacy was not a significant predictor of cancer prevention behaviour, improved information would most likely reduce a number of these misconceptions.

Another important finding in this study was the complexity associated with cancer prevention communications. Although some participants clearly asked for more information, others indicated being overwhelmed by the sheer amount of it. The cancer information overload (CIO) phenomenon has been described as impacting as much as 75% of adults in surveys [34,35].

The participants in our study raised a number of solutions for how to improve cancer prevention literacy through effective communication. For instance, culture-specific information delivered by trusted sources and the use of various communication methods, as well as the provision of assistance when navigating information. Some of these suggestions are in line with the results of a previous review by Fang and Ragin [36], who also concluded that navigation assistance is needed to overcome the barriers related to cancer screening among U.S. immigrants. Furthermore, Sørensen et al. also [37] highlight the importance of navigational support in health-care systems for individuals, which is crucial for improved cancer literacy. The current study suggests that this type of navigational support should be provided by families, friends, peer advisors, health-care providers, supporting staff or similar resources. Previous research among people with IDs supports this [23]. De Jesus et al., 2021 [13] also concluded, in their interview study on cancer screening among women with immigrant backgrounds in France, that better targeted and audience-responsive cancer prevention communication is required.

In the current study, we discovered variations in the sources of information that were believed and trusted. Some of the younger participants described themselves as trusting influencers and/or networks on social media, whereas others described a trust mainly for professionals, such as school nurses, teachers and physicians. We also found differences in the preferred channels of information, with some participants preferring information via social media, while others favoured face-to-face-based communication. There are several reasons for these varied preferences, on both an individual and a group level. Differences in health-information-seeking behaviours may be associated with health literacy. Sundell et al. [20] found in their study that individuals with low health literacy were less likely to seek the digital health information provided by the Swedish national health service.

We found several examples of general and subgroup-specific vulnerabilities, which seemingly impact cancer prevention literacy. For instance, the inability to comply with some of the ECAC recommendations among participants with IDs due to inadequate support from the staff at their homes and workplaces. This highlights the importance of supportive societal structures in addition to individual actions. This could include facilitating factors, such as the clear labelling of food and firmer bans and regulations on tobacco and alcohol, as well as easy-access vaccination and screening programmes. Sørensen et al. [37] concluded that cancer literacy (including primary, secondary and tertiary cancer prevention) could be improved by empowering citizens, using a combination of individual support and policy structures (such as National Cancer Control Plans and Europe’s Beating Cancer Plan). Even though it is stated in the ECAC that the recommendations need to be supported by governmental policies and actions, these policies and actions are not always in place, known or accessible. Espina [38] published a paper with eight recommendations on how to improve future editions of the ECAC. In the current study, we found indications supporting most of these recommendations. For example, the provision of adapted messages for the different target groups.

One of the strengths of this study is the rich data from four different and scarcely studied population subgroups. The study provides in-depth explanations about cancer prevention literacy among these groups, which are difficult to find using a quantitative research design.

Using online FGIs for data collection helped to facilitate both the accessibility and the interactions among the participants, who live in 13 different European countries (especially given that the interviews were conducted during the COVID-19 pandemic). For the FGIs among people with IDs, adaptations were made (interviews were held in the participant’s native language with experienced and well-known supportive staff present) to mitigate the interviews. Another strength was the use of a well-established analytic method (FA), which was suitable for qualitative research using these types of data and contributed to trustworthiness and rigor.

The FGI amongst people with immigrant backgrounds (subgroup B, FGI 3) were differently composed, compared with the other FGIs. This may have impacted the results. All participants in this FGI came from immigrant backgrounds, but they also acted as peer advisors. From this perspective, these participants probably had a higher level of cancer prevention literacy (as they have undergone an education program in their roles as peer advisors). On the other hand, these participants provided us with both unique insights and experiences from their interactions with various immigrant groups. One other possible limitation is that we did not discuss the results with representatives of the subgroups to verify the interpretations.

## 5. Conclusions

To improve cancer prevention literacy in Europe, greater attention is required to understand and overcome the barriers amongst different population subgroups. Based on the results in this current study, the authors recommend the following actions: First, improved and adapted cancer prevention information is needed to raise awareness and reduce the risk of misconceptions about cancer. Second, greater support for the individuals within the different subgroups (families, friends, peers, health-care providers, supportive staff, etc.) is also required. In addition, improved supportive societal structures are required. For instance, stringent regulations on smoking bans that includes SHS; affordable, healthy food; and easy-access vaccination and cancer-screening programmes. The results from this study could contribute to the implementation of Europe’s Beating Cancer Plan, which aims to make at least 80% of the European population aware of the code by 2025.

More research is needed to evaluate cancer prevention actions amongst different population subgroups.

## Figures and Tables

**Figure 1 ijerph-20-05888-f001:**
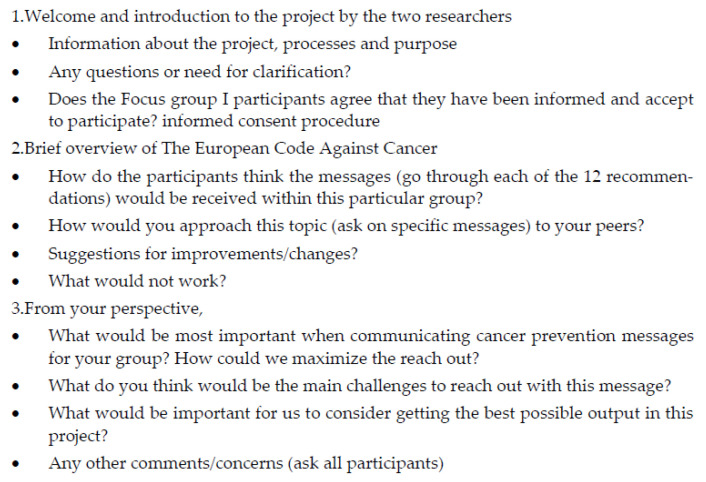
Interview guide.

**Figure 2 ijerph-20-05888-f002:**
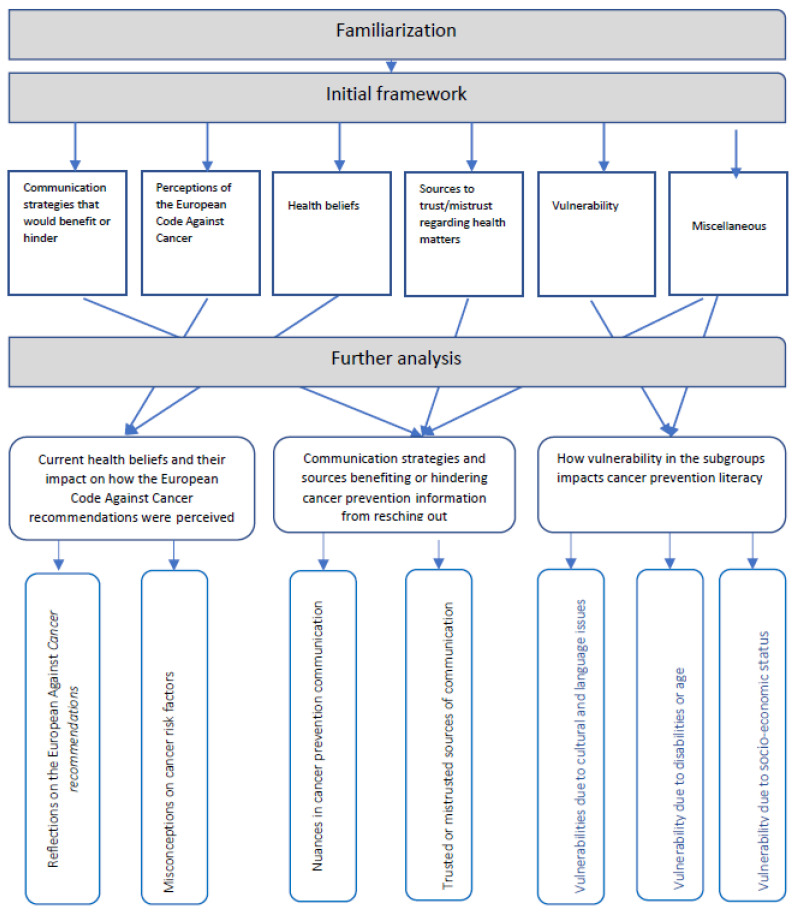
The analytic process according to framework analysis (FA).

**Table 1 ijerph-20-05888-t001:** Overview of the focus groups.

Description of Group	Participants n	Participating Countries	Gender	Language	Assessors Present	Additional Support People Present	Transcript, Number of Pages	Duration, Minutes
*Subgroup A* Young people (aged 18–29) without personal cancer experience (focus group 1)	2	Sweden	1 woman 1 man	English	2	0	30	71
*Subgroup A* Young people (aged 18–29) without personal cancer experience (focus group 2)	8	Croatia Georgia Czech RepublicTurkey	4 women 4 men	English	2	0	35	107
*Subgroup B* People with an immigrant background (focus group 3)	8	Sweden	6 women 2 men	Swedish	1	0	30	99
*Subgroup C* Young people (aged 18–29) with personal cancer experience (focus group 4)	8	Romania The Netherlands Portugal Poland Macedonia Serbia The Republic of Moldova Turkey	4 women 4 men	English	1	0	31	99
*Subgroup D* People with intellectual disabilities (IDs) (focus group 5)	6	Sweden	4 women 2 men	Swedish	1	1 personal assistant, 2 coordinators from support group	37	104
*Subgroup D* People with IDs (focus group 6)	8	United Kingdom	4 women 4 men	English	1	2 coordinators from support group	35	94
*Total*	40		23 women 17 men			5	198	574

## Data Availability

The data presented in this study are available on request from the corresponding author. The data are not publicly available due to confidentiality reasons.

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
