# Peer review of "Cancer Prevention Literacy among Different Population Subgroups: Challenges and Enabling Factors for Adopting and Complying with Cancer Prevention Recommendations"

_ijerph, 2023, doi:10.3390/ijerph20105888_

Round 1

Reviewer 1 Report

This paper presents very interesting findings not investigated before, on understanding barriers to grasping the messages in the European Code Against Cancer (ECAC) by immigrant or marginalised persons.  While one may need to be cautious of the findings due to the sample size of n=40, this paper provides crucial insights on possible reasons why the ECAC does not seem to be reaching certain populations.  Perhaps it would be worthwhile to point out that the findings can be helpful in supporting the EU's Beating Cancer Plan, which "aims to make at least 80% of the population aware of the Code by 2025."  https://health.ec.europa.eu/system/files/2022-02/eu_cancer-plan_en_0.pdf What would be also helpful (but not critical) is a summary or bullet points of recommendations by the authors on how to better communicate and raise awareness of ECAC messages among marginalised populations.

Author Response

Thank you for the positive comments. As suggested, we have added the following sentence in the conclusion (p.16) The results from this study could contribute to the implementation of EU’s Beating Cancer Plan, which aims to make at least 80% of the European population aware of the Code by 2025.

We have considered the suggestion to make an additional summary or bullet points with recommendations, as suggested. However, we believe that we already have included, in both the abstract and the conclusions, three general recommendations (that improved and adapted cancer prevention information is needed, as well as greater support for individuals in addition to improved societal structures). We also give a few examples here. We find it difficult to be more specific as these recommendations need to be adapted to different contexts, which we also discuss in the Discussion section. We have however tried to clarify that the authors recommend these actions in the conclusion section by adding the following sentence: Based on the results in this current study, the authors recommend the following actions;

Reviewer 2 Report

Thank you for the opportunity to review your manuscript.

General comments: The topic is very relevant and the methods are very well described. If the journal does not explicitly request it, I would recommend moving away from a passive voice and using an active voice to describe your methods and findings:  "We reviewed the data" instead of "data was revised" 

I recommend adding the 12 ECAC recommendations as an appendix. 

a. Abstract: Clear recount of the paper.

1. The last phrase "supporting societal" is vague and not very clear. Could you explain further or re-phrase? 

b. Introduction: Clear and well written. Need to provide some references for some of the most relevant statements. 

Please provide references for these statements: 

1. "It is estimated in Europe’s Beating Cancer Plan that 40 % of the cancer cases in Europe are preventable"

2. "Without these actions, the cancer mortality within the European Union (EU) is expected to increase by more than 24% by 2035, making it the leading cause of death across the continent"

3. "Cancer prevention can be improved dramatically by raising awareness and addressing the risk factors, such as tobacco and alcohol consumption, lack of physical activity, obesity, unhealthy diet, extensive sun exposure, and exposure to pollution"

4. A more commonly used word for "success factors" would be "facilitators" 

5. There are two paragraphs that are too similar to others found on the internet at cancercentrum.se

i. Cancer prevention can be improved dramatically by raising awareness and addressing the risk factors, such as tobacco and alcohol consumption, lack of physical activity, obesity, unhealthy diet, extensive sun exposure, and exposure to pollution.

ii. Inequalities in survival rates and other cancer outcomes have been reported frequently during the last decades, both within and between countries. It is well documented that cancer risk factors are impacted by socio-economic factors [8];[9] [10];[11];[12];[13]. Research has shown that socio-economic inequities impact exposure to risk factors, access to screening and other preventive services, as well as diagnostics, treatment and even palliative care. 

I recommend re-phrasing or citing to avoid plagiarism. 

c. Materials and methods

1. The description of the methods is very exhaustive.  Did the three authors perform all stages of analysis? Please clarify.  I suggest reviewing the COREQ checklist for reporting qualitative research. In it, there are a few points being made to ensure transparency and comprehensiveness of the report, such as describing some of the authors' characteristics, theoretical frameworks used (if any), and description of the analysis. (How many coders? Who?)

https://academic.oup.com/intqhc/article/19/6/349/1791966

2.  Figure 2 appears before Figure 1 in the manuscript. Please, re-label.  Also, it needs to be reformatted as it has duplicated parts.  

3. Sub-group B; People with an immigrant background: Typo: the coordinator of this network (HOW had no involvement...)

4. Committee for Ethical and Professional issues at the Clinical Hospital Center Osijek, Croatia in February 2022 (R1-1968-3/2022) is written in a different font. 

5. The paragraph:  Communication strategies that would benefit or hinder reaching out; Perceptions of the ECAC recommendations; Health Beliefs; Sources to trust/mistrust regarding health matters; Vulnerability and Miscellaneous (where potentially interested findings were listed that didn’t fit into any of the other initial themes) id written in a different font. 

6. How did you make sure people with intellectual disabilities understood your questions?

7. Did you consider sharing the analysis with any of the representatives of the sub-groups to ensure the correct interpretation of the results? 

d. Results:

1. Reformat Table 1. I suggest removing information on the number of transcript pages and adding information on the mean age for each sub-group. 

2. I suggest labeling the categories and subcategories to give the reader a clearer understanding of how they relate to each other. For example:

I. Current health beliefs and their impact on how the European Code Against Cancer (ECAC) recommendations were perceived.

I.a. Reflections on the European Code Against Cancer (ECAC) recommendations

I.b Misconceptions on cancer risk factors. 

3. Consider making a table of quotes or adding more quotes to the categories to make sure the participants' voice is well represented. 

4. The phrase: " Another example of cultural vulnerability raised by the immigrant group, was having to exclude yourself from important social events if striving for a healthy lifestyle. For instance, events including unhealthy foods and/or water pipe smoking" is in a different font.

e. Discussion: The solutions raised by the participants are very interesting and get lost in the text. Maybe put them in a Table/Figure/Box to highlight them.

Reviewer 3 Report

This paper describes the results of qualitative focus group interviews with different populations to understand perceptions of cancer prevention guidelines and strategies for communication about cancer prevention. The results are important, but are wide ranging and could be described more clearly with and perhaps organized more efficiently  with headings/subheadings. The groups are also very differebt and so it is hard to synthesize the information, for example, young cancer survivors and people with intellectual disabilities are all included in the study. It may have been better to focus on one population at a time and recruit more people within each group. The discussion would benefit from clear takeaways, an dindicate how this study goes beyond what was published in this reference on page 16  Espina [39] published a paper with eight recommendations on how to improve future editions of the ECAC. In the current study, we found indications sup-porting most of these recommendations.”

Please list the 12 recommendations from the ECAC which were shown to participants.

Please discuss: Are the ECAC recommendations meant to be viewed by the public as a list –or are they recommendations for health care providers, public health practitioners and researchers etc. to develop communication strategies for the public/patients? That is, is the ECAC meant to have a general public audience, was it designed with health communication to the general public in mind?

Figure 2 is referenced before Figure 1. Figure 2 Interview guide appears repeated in text, as a bulleted list and is not formatted correctly. Acronyms FDG and FGI are hard to follow, could these be simplified and spelled out for clarity for the reader?

Can you discuss the rationale for participant recruiting? For example, why have cancer nurses recruit young people without cancer experience from their personal networks (knowing a cancer nurse may make these people more familiar with cancer experiences)?

Sub group B-please clarify whether the peer advisors were participants or recruited participants?

Was there any attempt to correct misconcpetions shared within the groups?(e.g. p11 blood sugar/alcohol discussion)

Headings and subheading should be made distinct—it is unclear what is a subheading in the results section.

Round 2

Reviewer 3 Report

Thank you, the responses you provided should also be included in a revised version of the manuscript. 

Author Response

Thank you for the useful feedback, copied in below. After each comment, we have inserted our response (in italics).

This paper describes the results of qualitative focus group interviews with different populations to understand perceptions of cancer prevention guidelines and strategies for communication about cancer prevention. The results are important, but are wide ranging and could be described more clearly with and perhaps organized more efficiently with headings/subheadings. The groups are also very different and so it is hard to synthesize the information, for example, young cancer survivors and people with intellectual disabilities are all included in the study. It may have been better to focus on one population at a time and recruit more people within each group. The discussion would benefit from clear takeaways, and indicate how this study goes beyond what was published in this reference on page 16  “Espina [39] published a paper with eight recommendations on how to improve future editions of the ECAC. In the current study, we found indications sup-porting most of these recommendations.”

Yes, you are correct that the current study includes participants from different population sub-groups. We believe that we do present results, both from each of the sub-group’s perspective, but also some findings were similar and occur in more than one or all of the studied sub-groups. The aim was to gain knowledge and deep understanding on cancer prevention literacy from the perspective of the four population sub-groups (hence the choice of a qualitative design), rather than to collect data from large groups, to maximize generalizability. We agree that more research is needed to understand the perspective in different sub-groups, which we also suggest in the conclusion section.

We thank you for notifying how we refer to the paper by Espina et al 2021. Unfortunately, we have cited the wrong publication here, and apologize for that. The correct reference should be Espina et al 2021, Sustainability and monitoring of the European Code Against Cancer: Recommendations Cancer Epidemiology 72 (2021) 101933, which we have now corrected accordingly. In reference to these recommendations by Espina et al 2021, we state: In the current study, we found indications supporting most of these recommendations. For example, provision of adapted messages for the different target groups (p.16).

In addition, we believe our results provide in-depth explanations on cancer prevention literacy among these groups, which is not the purpose of the paper by Espina et al.

Please list the 12 recommendations from the ECAC which were shown to participants.

 A similar request was raised by reviewer 2. We agree that it would be good to facilitate for the reader but suggest instead to insert a web link to the ECAC page, as the recommendations are clearly described there in different languages together with the scientific justifications (the link has been inserted in the revised manuscript, p.2). However, if the reviewer insists, we are happy to provide a list in an appendix.

Please discuss: Are the ECAC recommendations meant to be viewed by the public as a list –or are they recommendations for health care providers, public health practitioners and researchers etc. to develop communication strategies for the public/patients? That is, is the ECAC meant to have a general public audience, was it designed with health communication to the general public in mind?

The ECAC is an initiative to inform people about actions they can take for themselves or their families to reduce their risk of cancer. The target population is the general public and it has been developed by groups of medical experts in collaboration with experts in behavioural research and communication. The 4th edition consists of twelve recommendations that most people can follow without any special skills or advice. It focuses on actions that individual citizens can take to help prevent cancer. Successful cancer prevention requires these individual actions to be supported by governmental policies and actions. The participants in our focus groups, as discussed on p.15, expressed that the recommendations would reach a larger audience and make a better impact, if there was a clearer focus on the health benefits and gains.

Figure 2 is referenced before Figure 1. Figure 2 Interview guide appears repeated in text, as a bulleted list and is not formatted correctly.

Thank you for highlighting this. We have corrected this accordingly in the revised manuscript (p. 4 and 8).

 Acronyms FDG and FGI are hard to follow, could these be simplified and spelled out for clarity for the reader?

In the revised manuscript, we only use Focus Group Interview (FGI) or Focus group (spelled out) for clarity.

Can you discuss the rationale for participant recruiting? For example, why have cancer nurses recruited young people without cancer experience from their personal networks (knowing a cancer nurse may make these people more familiar with cancer experiences)?

We believe that we do explain in the background section (p.2-3) why we selected these four sup-groups. For the focus group among young people without own experience of cancer, we chose to engage the network of young cancer nurses within European Oncology Nursing Society, as they could approach people in their personal networks, and with their specific expertise, explain the study process and aim in detail, when needed. Also, this network helped us maintain confidentiality, as the young cancer nurses facilitated the contacts, including meeting invites/links. We also wanted to recruit participants from different countries in Europe, both men and women, which also the young cancer nurses could facilitate. This sub-group could have been recruited via some other organization, or via online recruitment. We believe however, that the process we used was appropriate, and we have not identified any signs that these participants were more familiar with cancer prevention issues, as suggested.

Sub group B-please clarify whether the peer advisors were participants or recruited participants?

The peer advisors were participants. We describe this on p. 5 (The peer advisors participating in this FGI…) and discuss that these participants had a different role (p. 17… All participants in this FGI came from immigrant backgrounds, but they also acted as peer advisors…).

Was there any attempt to correct misconceptions shared within the groups?(e.g. p11 blood sugar/alcohol discussion)

We have contacted to coordinator for the peer advisors after the FGI and shared our experiences. She has in turn, contacted the participants to clarify the misconceptions mentioned by the reviewer.

Headings and subheading should be made distinct—it is unclear what is a subheading in the results section.

The heading and subheadings are following the journals template, but we have in the revised version added space between each paragraph, for clarity.

In addition, the revised manuscript has been reviewed by a professional proof-reader from Edinburgh Napier University.
